# Machine Explanations and Human Understanding

**Chacha Chen\***                                                        *chacha@uchicago.edu*
*Department of Computer Science*
*University of Chicago*

**Shi Feng\***                                                              *shif@uchicago.edu*
*Department of Computer Science*
*University of Chicago*

**Amit Sharma**                                                       *amshar@microsoft.com*
*Microsoft Research*

**Chenhao Tan**                                                      *chenhao@uchicago.edu*
*Department of Computer Science*
*University of Chicago*

**Reviewed on OpenReview:** *https:// openreview. net/ forum? id=y4CGF1A8VG*

## Abstract

Explanations are hypothesized to improve human understanding of machine learning models and achieve a variety of desirable outcomes, ranging from model debugging to enhancing human decision making. However, empirical studies have found mixed and even negative results. An open question, therefore, is *under what conditions explanations can improve human understanding and in what way*. To address this question, we first identify three *core* concepts that cover most existing quantitative measures of understanding: *task decision boundary*, *model decision boundary*, and *model error*. Using adapted causal diagrams, we provide a formal characterization of the relationship between these concepts and human approximations (i.e., understanding) of them. The relationship varies by the level of human intuition in different task types, such as emulation and discovery, which are often ignored when building or evaluating explanation methods. Our key result is that human intuitions are necessary for generating and evaluating machine explanations in human-AI decision making: without assumptions about human intuitions, explanations may improve human understanding of model decision boundary, but *cannot* improve human understanding of task decision boundary or model error. To validate our theoretical claims, we conduct human subject studies to show the importance of human intuitions. Together with our theoretical contributions, we provide a new paradigm for designing behavioral studies towards a rigorous view of the role of machine explanations across different tasks of human-AI decision making.

## 1 Introduction

Despite the impressive performance of machine learning (ML) models (He et al., 2015; Mnih et al., 2015; Silver et al., 2017), there are growing concerns about the black-box nature of these models. Explanations are hypothesized to improve human understanding of these models and achieve a variety of desirable outcomes, ranging from helping model developers debug (Hong et al., 2020) to improving human decision making in critical societal domains (Green & Chen, 2019a;b; Lai & Tan, 2019; Lai et al., 2020; Poursabzi-Sangdeh et al., 2021; Zhang et al., 2020). However, the exact definition of human understanding remains unclear. Moreover,

---

*Equal contribution.

empirical experiments report mixed results on the effect of explanations (Adebayo et al., 2022; Green & Chen, 2019a;b; Lai & Tan, 2019; Poursabzi-Sangdeh et al., 2021; Ribeiro et al., 2016), raising an intriguing puzzle about what factors drive such mixed results. We argue that frameworks of how human understanding is shaped by machine explanations (Alvarez-Melis et al., 2021; Yang et al., 2022) is key to solving this puzzle. We prove that effective machine explanations are theoretically impossible without incorporating human intuitions.

**Roadmap and contributions.** We begin by tackling the question of what human understanding we would like to achieve in the context of human-AI decision making in classification tasks (Section 2). We identify three core concepts of interest from existing literature: 1) **task decision boundary** (*deriving the true label in the prediction problem, the key target for decision making both for humans and models* (Bansal et al., 2020; Biran & McKeown, 2017; Buçinca et al., 2020; 2021; Carton et al., 2020; Doshi-Velez & Kim, 2017; Feng & Boyd-Graber, 2018; Gonzalez et al., 2020; Guo et al., 2019; Kiani et al., 2020; Lai & Tan, 2019; Lai et al., 2020; Liu et al., 2021; Nourani et al., 2021; Poursabzi-Sangdeh et al., 2021; Weerts et al., 2019; Zhang et al., 2020)), 2) **model decision boundary** (*simulating the model predicted label, evidence of strong human understanding of the model* (Alqaraawi et al., 2020; Buçinca et al., 2020; Chandrasekaran et al., 2018; Chromik et al., 2021; Doshi-Velez & Kim, 2017; Friedler et al., 2019; Hase & Bansal, 2020; Lakkaraju et al., 2016; Lipton, 2016; Liu et al., 2021; Lucic et al., 2020; Nguyen, 2018; Nourani et al., 2021; Poursabzi-Sangdeh et al., 2021; Ribeiro et al., 2018; Wang & Yin, 2021)), and 3) **model error** (*recognizing whether a predicted label is wrong, a useful intermediate variable in decision making and a central subject in trust* (Bansal et al., 2019; 2020; Biran & McKeown, 2017; Buçinca et al., 2020; 2021; Bussone et al., 2015; Carton et al., 2020; Feng & Boyd-Graber, 2018; Gonzalez et al., 2020; Kiani et al., 2020; Lai & Tan, 2019; Lai et al., 2020; Liu et al., 2021; Nourani et al., 2021; Poursabzi-Sangdeh et al., 2021; Wang & Yin, 2021; Weerts et al., 2019; Zhang et al., 2020)

To enable a rigorous discussion of how machine explanations can shape human understanding, we develop a theoretical framework of human understanding with adapted causal diagrams (Section 3). Depending upon what is shown to the person (e.g., the model's prediction on an instance), different causal relationships emerge among the core variables and their human approximations. To formalize these relationships, we introduce an operator, *show*, that articulates how assumptions/interventions (henceforth conditions) shape human understanding. We consider two conditions: 1) whether a person has the perfect knowledge of the task and 2) whether machine predicted labels are revealed. Going through these conditions yields a tree of different scenarios for human understanding (Section 4).

We then incorporate machine explanations in the framework to vet the utility of explanations in improving human understanding (Section 5). Our causal diagrams reveal the critical role of *task-specific* human intuitions[1] in effectively making sense of explanations, and shed light on the mixed findings of empirical research on explanations. We first point out that existing explanations are mostly derived from model decision boundary. Although explanations can potentially improve human understanding of model decision boundary, they *cannot* improve human understanding of task decision boundary or model error without assumptions about task-specific intuitions. In other words, *complementary performance (i.e., human+AI > human; human+AI > AI) is impossible without assumptions about task-specific human intuitions.* To achieve complementary performance, we articulate possible ways that human intuitions can work together with explanations.

Finally, we validate the importance of human intuitions through human-subject studies (Section 6). To allow for full control of human intuitions, we use a Wizard-of-Oz setup (Dahlbäck et al., 1993). Our experimental results show: 1) when we remove human intuitions, humans are more likely to agree with predicted labels; 2) participants are more likely to agree with the predicted label when the explanation is consistent with their intuitions. These findings highlight the importance of articulating and measuring human intuitions.

Our key theoretical results emphasize the importance of articulating and measuring human intuition in order to develop machine explanations that enhance human understanding. Our experiments act as a minimal viable product, demonstrating how we can make assumptions on human task-specific intuitions to predict when and in what way explanations can be helpful. Note that our experiment results do not directly substantiate our primary theorems, but they do provide a strong supporting evidence, showing a promising way forward in

---

[1]For example, which feature is important for the decision; see full definition in Section 3.2.

this emerging field of human-AI interactions. Altogether, our work contributes to the goal of developing a rigorous science of explanations (Doshi-Velez & Kim, 2017).

## 2 Three Core Concepts for Measuring Human Understanding

In this section, we identify three key concepts of interest for measuring human understanding in human-AI decision making: task decision boundary, model decision boundary, and model error. We present high-level definitions of these concepts and formalize them in Section 3.[2]

We use a two-dimensional binary classification problem to illustrate the three concepts (Figure 1). *Task decision boundary*, as represented by the dashed line, defines the mapping from inputs to ground-truth labels: inputs on the left are positive and the ones on the right are negative. *Model decision boundary*, as represented by the solid line, determines model predictions. Consequently, the area between the two boundaries is where the model makes mistakes. This yellow highlighted background captures *model error*, i.e., where the model prediction is incorrect. With a perfect model, the model decision boundary would be an exact match of the task decision boundary, and model error is empty.[3]

To the best of our knowledge, we are not aware of any existing quantitative behavioral measure of human understanding that does not relate to one of these three concepts of interest.[4] Building on a recent survey (Lai et al., 2021), we identify 32 papers that: 1) use machine learning models and explanations with the goal of improving human understanding; and 2) conduct empirical human studies to evaluate human understanding with quantitative metrics. Although human-subject experiments can vary in subtle details, the three concepts allow us to organize existing work into congruent categories. We provide a reinterpretation of existing behavioral measures using the three concepts below (a detailed summary of the papers is in Appendix B)[5].

$- - -\ f(\cdot)$    $———\ g(\cdot)$    $z(\cdot)$
**Task decision boundary**    **Model decision boundary**    **Model error**

Figure 1: Illustration of the three core concepts using a binary classification problem. Task decision boundary (dashed line) defines the ground-truth mapping from inputs to labels. Model decision boundary (solid line) defines the model predictions. Model error (highlighted area) represents where the model's predictions are incorrect.

**Measuring human understanding of task decision boundary (and model error) via human+AI performance.** Similar to the application-grounded evaluation defined in Doshi-Velez & Kim (2017), the most well-adopted evaluation measurement of human understanding is to measure human understanding of the task decision boundary through human+AI performance (Bansal et al., 2020; Biran & McKeown, 2017; Buçinca et al., 2020; 2021; Carton et al., 2020; Doshi-Velez & Kim, 2017; Feng & Boyd-Graber, 2018; Gonzalez et al., 2020; Guo et al., 2019; Kiani et al., 2020; Lai & Tan, 2019; Lai et al., 2020; Liu et al., 2021; Nourani et al., 2021; Poursabzi-Sangdeh et al., 2021; Weerts et al., 2019; Zhang et al., 2020). In those experiments, participants are presented with machine predictions and explanations, then they are asked to give a final decision based on the information, with the goal of achieving complementary performance. For example, human decision-makers are asked to predict whether this defendant would re-offend within two years, given a machine prediction and explanations (Wang & Yin, 2021). For binary classification problems, measuring human understanding of the model error is equivalent to measuring human understanding of the task decision boundary when machine predictions are shown.

**Measuring human understanding of model decision boundary via human simulatability.** A straightforward way of evaluating human understanding of model decision boundary is to measure how

---

[2]In this work, we omit subjective measures such as cognitive load and subjective assessments of explanation usefulness.

[3]We present a deterministic example for ease of understanding and one can interpret this work with deterministic functions in mind. In general, one can also think of model decision boundary, task decision boundary, and model error probabilistically.

[4]As we will show later, understanding the mechanism behind a model's predictions (e.g., feature importance) falls under the category of understanding the model decision boundary.

[5]Also available at `https://github.com/Chacha-Chen/Explanations-Human-Studies`.

well humans can simulate the model predictions, or in other words, the human ability of forward simulation/prediction (Doshi-Velez & Kim, 2017). Humans are typically asked to simulate model predictions given an input and some explanations (Alqaraawi et al., 2020; Buçinca et al., 2020; Chandrasekaran et al., 2018; Chromik et al., 2021; Doshi-Velez & Kim, 2017; Friedler et al., 2019; Hase & Bansal, 2020; Lakkaraju et al., 2016; Lipton, 2016; Liu et al., 2021; Lucic et al., 2020; Nguyen, 2018; Nourani et al., 2021; Poursabzi-Sangdeh et al., 2021; Ribeiro et al., 2018; Wang & Yin, 2021). For example, given profiles of criminal defendants and machine explanations, participants are asked to guess what the AI model would predict (Wang & Yin, 2021).

**Measuring human understanding of model decision boundary via counterfactual reasoning.** Sometimes researchers measure human understanding of the decision boundary by evaluating participants' counterfactual reasoning abilities (Friedler et al., 2019; Lucic et al., 2020). Counterfactual reasoning investigates the ability to answer the 'what if' question. In practice, participants are asked to determine the output of a perturbed input applied to the same ML model (Friedler et al., 2019). Lucic et al. (2020) asked participants to manipulate the input to change the model output.

**Measuring human understanding of model decision boundary via feature importance.** Additionally, Wang & Yin (2021) also tested human understanding of model decision boundary via feature importance, specifically by (1) asking the participants to select among a list of features which one was most/least influential on the model's predictions and (2) specifying a feature's marginal effect on predictions. Ribeiro et al. (2016) asked participants to perform feature engineering by identifying features to remove, given the LIME explanations. These can be viewed as a coarse inquiry into properties of the model's model decision boundary.

**Measuring human understanding of model error through human trust.** In some other cases, trust or reliance is introduced as a criterion reflecting human understanding of model error. Explanations are used to guide people to trust an AI model when it is right and not to trust it when it is wrong. Hence, by analyzing when and how often human follows machine predictions, trust can reflect the human understanding of the model error (Buçinca et al., 2021; Wang & Yin, 2021; Zhang et al., 2020). In other cases, the measure of human understanding of model error can be used as an intermediate measurement towards measuring task decision boundary (Bansal et al., 2019; 2020; Biran & McKeown, 2017; Buçinca et al., 2020; 2021; Bussone et al., 2015; Carton et al., 2020; Feng & Boyd-Graber, 2018; Gonzalez et al., 2020; Kiani et al., 2020; Lai & Tan, 2019; Lai et al., 2020; Liu et al., 2021; Nourani et al., 2021; Poursabzi-Sangdeh et al., 2021; Wang & Yin, 2021; Weerts et al., 2019; Zhang et al., 2020), where human subjects are asked whether they agree with machine predictions.

## 3 A Theoretical Framework of Human Understanding

We introduce a theoretical framework to formalize human understanding of the three core concepts in the context of human-AI decision making. This framework enables a rigorous discussion on human understanding as well as the underlying assumptions/interventions that shape those understanding, and sets up our discussion about the role of machine explanations.

### 3.1 Core Functions

Formally, the three core concepts are functions defined w.r.t. a prediction problem and a ML model:

- **Task decision boundary** is a function $f : \mathbb{X} \to \mathbb{Y}$ that represents the groundtruth mapping from an input $X$ to the output $Y$.

- **Model decision boundary** is another function $g : \mathbb{X} \to \mathbb{Y}$ that represents our ML model and outputs a prediction $\hat{Y}$ given an input. $g$ is usually trained to be an approximation of $f$. We assume that we are given a trained model $g$; the training process of $g$ is not crucial for this work. Another important assumption is that $\hat{Y}$ is the best prediction that $g$ can provide for $X$. In other words, $g$ does not contain information that can improve its prediction for $X$.

- **Model error** is an indicator of whether the model prediction differs from the groundtruth for an input: $z(X, f, g) = \mathbb{I}[f(X) \neq g(X)], \forall X \in \mathbb{X}$. We use $z(X)$ for short when the omitted arguments $f$ and $g$ are clear from context.

We call them *core* functions as they underpin human understanding.

Our work is concerned with a test instance $X$ that is *independent* from the training data and the given model $g$.[6] Given $X$, we refer to the outputs of core functions $Y$, $\hat{Y}$, and $Z$ as the three local core variables. Note that the core functions do not involve any people; they exist even in the absence of human understanding.

### 3.2 Human Approximation (Understanding) of the Core Functions

We use $H$ to represent **task-specific human intuitions**.[7] For example, in prostate cancer diagnosis, $H$ may include the knowledge of the location of prostate and the association between darkness and tumor; in sentiment analysis, $H$ can refer to the ability to understand the semantics of language and interpret its polarity; in recidivism prediction, $H$ may capture the intuition that older people tend to not recidivate. We define $H$ as human intuitions that are relevant to the task at hand and distinct from more general intuitions, such as the ability to distinguish different colors in interpreting feature importance. Note that $H$ may not always be valid.

We then use $f^H$, $g^H$, and $z^H$ to denote the human's *subjective* approximations of the core functions, each of them being a function with the same domain and codomain as its objective counterpart. These human approximations can be interpreted as mental models, influenced by the human intuitions ($H$), which can change over time as the human-AI interaction progresses.

Table 1 summarizes the notations for core functions and human understandings.

Given the definition of these concepts, We can rephrase common cooperative tasks in human-AI decision making in terms of the core functions and human understanding grouped by stakeholders:

- For **decision makers** such as doctors, judges, and loan officers, the main goal is to improve their understanding of task decision boundary ($f^H$).

- For **decision subjects** such as patients, defendants, and loan applicants, the object of interest can differ even for these three examples. Patients care about the task decision boundary more, while defendants and loan applicants may care about the model decision boundary and especially model error, and would like to figure out how they can appeal model decisions.

- **Model developers** might be most interested in model error, and the eventual goal is to change the model decision boundary.

- For **algorithm auditors**, the main goal is to figure out whether the model decision boundary and model error conform to laws/regulations.

**Measuring human understanding in empirical studies.** The distance between core functions and their human approximations can be used as a measure for human understanding. Since human approximations are theoretical constructs that only exist in the human brain, we need to perform user studies to measure them. For example, we can ask a human to guess what the model would have predicted for a given input $X$; the human's answer $\hat{Y}^H$ characterizes their *local* understanding of the model decision boundary. In the rest of the paper, one can interpret "human understanding" with this particular measurement of human approximations. Perfect human understanding thus refers to 100% accuracy in such measurement.

---

[6]Training data and $g$ are not independent.

[7]Clarification on our choice of the word, **intuition**: (1) we use intuition instead of belief, because intuition is broader than belief. For example, knowledge is typically not counted as belief; (2) we use intuition instead of prior, since we do not emphasize the difference between prior and posterior. In fact, sometimes intuitions can be activated because of showing explanations, so it is certainly not prior in that case.

| Variable | Function | Description |
|---|---|---|
| X | — | Input instance |
| $Y$ | $f : \mathbb{X} \to \mathbb{Y}$ | Task decision boundary |
| $\hat{Y}$ | $g : \mathbb{X} \to \mathbb{Y}$ | Model decision boundary |
| $Z$ | $z : \mathbb{X} \to \mathbb{Z}$ | Model error |
| $Y^H$ | $f^H : \mathbb{X} \to \mathbb{Y}$ | Human understanding of the $f$ |
| $\hat{Y}^H$ | $g^H : \mathbb{X} \to \mathbb{Y}$ | Human understanding of the $g$ |
| $Z^H$ | $z^H : \mathbb{X} \to \mathbb{Z}$ | Human understanding of the $z$ |
| $H$ | — | Task-specific human intuitions |
| $E$ | $e(g) : \mathbb{X} \to \mathbb{E}$ | Machine explanations |

Table 1: A summary of notations.

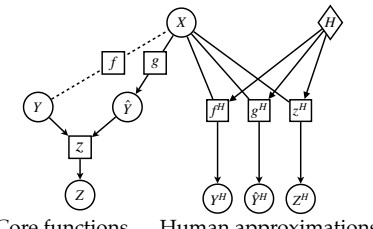

Core functions     Human approximations

Figure 2: Visualizing the relations between local variables, core functions, and human approximations of them.

We assume that the human approximation remains static and examine a human's local understanding with our framework in the main paper. We note that improving human global understanding is often the actual goal in many applications, and add discussions on global understanding in Appendix D.

## 3.3 Causal Graph Framework for Core functions

To reason about human understanding, we need to understand how core functions relate to each other, and how interventions may affect human understanding. To do so, we adapt causal directed acyclic graphs (DAGs) to formalize a causal model for the core functions of human understanding. We explain in more details about our adaptations of the causal diagrams in Appendix C.

Let us first look at core functions on the left in Figure 2. As our work is concerned with the causal mechanisms between variables, different from standard causal diagrams, we make them explicit by adding a functional node ($g$ in a square) on the edge from $X$ to $\hat{Y}$, indicating that $g$ controls the causal link from $X$ to $\hat{Y}$. $g$ is thus a parent of $\hat{Y}$. $X$ and $Y$ are connected with a dashed line through $f$ since we do not assume the causal direction between them (Veitch et al., 2021). $Z$ is the binary indicator of whether $Y$ and $\hat{Y}$ are different. From this part of the figure, we have the following observations:

- According to d-separation (Pearl, 2009), $\hat{Y}$ is independent of $Y$ given $X$ and $g$.

- $Z$ is a collider for $Y$ and $\hat{Y}$, so knowing $Z$ and $\hat{Y}$ entails $Y$ in binary classification.

On the right in Figure 2, We formulate human understanding of the core functions. Task-specific intuitions $H$ drives human mental models of the core functions. As this node generates functions, we use $\diamond$ to highlight its difference from other nodes. Figure 2 shows a base version of how human intuitions relate to human understanding of core variables. For now, we do not make any additional assumptions; we simply connect human intuition ($H$) with their understanding ($f^H, g^H, z^H$) and omit any potential connections between them. In other words, because $H$ has no connection with $f$, $g$, and $z$, $f^H, g^H, z^H$ are independent from $f, g, z$. Later, we will discuss more realistic instantiations on their relationships in Section 4.

According to d-separation, Figure 2 suggests that human approximation of core variables are independent from core variables given $X$ and $H$, without extra assumptions about human intuitions. Therefore, our goal is to articulate what assumptions we make and how they can improve human understanding.

**A new *show* operator.** Without extra assumptions, the causal direction between $Y^H, \hat{Y}^H, Z^H$ is unclear. We visualize this ambiguity by connecting nodes with undirected dashed links in Figure 3(a) as the base diagram. This base diagram is not useful in its current state; in order to use the diagram to reason about human understanding, we need *realizations* of the base diagram where dashed links are replaced by solid, directional links.

To disambiguate the dashed links in the base diagram, we need to articulate how assumptions and interventions can affect human understanding, i.e., $Y^H, \hat{Y}^H, Z^H$ and their corresponding local variables $Y, \hat{Y}, Z$. Thus, we introduce a new operator, *show*, to capture interventions on nodes and edges in causal diagrams. When *show* is applied to a core variable, that information becomes available to the human. For example, $show(\hat{Y})$ means that the human can see the model prediction $\hat{Y}$. This operation draws an equivalence between the

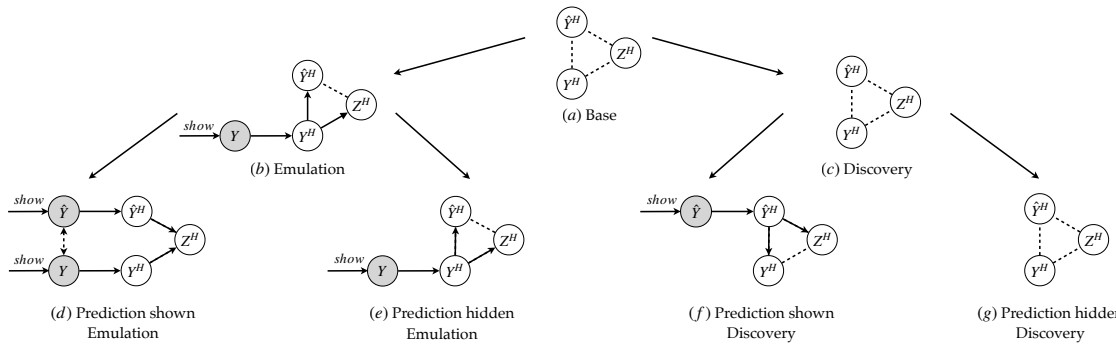

Figure 3: Causal diagrams visualizing the relationship between a human's local understanding ($Y^H, \hat{Y}^H, Z^H$). With the base diagram at the root, we organize its realizations based on different conditions in a two-level tree. Undirected dashed lines represent ambiguous causal links. Directed solid lines represent causal links. The bidirectional dashed line in subfigure (d) represents the correlation between $\hat{Y}$ and $Y$ potentially induced by the prediction model. Shaded nodes and their edges represent *show* operations.

core variable and the human approximated counterpart, i.e., $\hat{Y}^H = \hat{Y}$. That is, $show(\hat{Y})$ adds a link from $\hat{Y}$ to $\hat{Y}^H$ and removes all other edges going into $\hat{Y}^H$, effectively disambiguating the relation between $\hat{Y}^H$, $Y^H$ and $Z^H$ (Figure 3(f)).

We introduce the new *show* operator. As opposed to the standard *do* operator, the *show* operator can change the causal diagram as we reason about human understanding, including adding new edges and disambiguating dashed links. Notation-wise, *show* allows us to specify the condition for human approximations; e.g., $Y^H_{show(\hat{Y})}$ denotes the human local understanding of $Y$ given predicted label $\hat{Y}$. As a result, *show* provides strong flexibility in updating causal diagrams, and we believe that this flexibility provides an important first step to have a rigorous discussion on human understanding.

## 4 Human Understanding without Explanations

To illustrate the use of our framework, we start by analyzing human-AI decision making without the presence of explanations. We consider two conditions in Figure 3.

*Condition 1—emulation vs. discovery (Effect of $show(Y)$).* The first condition is an assumption about human knowledge, i.e., that the human has perfect knowledge about task decision boundary; in other words, $f^H$ perfectly matches $f$ and $Y^H = Y$ for all inputs (note that this is an assumption about $H$). Tasks where human labels are used as ground truth generally satisfy this condition, e.g., topic classification, reading comprehension, and object recognition. We follow Lai et al. (2020) and call them *emulation* tasks, in the sense that the model is designed to emulate humans; by contrast, discovery tasks are the ones humans do not have perfect knowledge of task decision boundary (e.g., deceptive review detection and recidivism prediction).[8]

Figure 3b and c illustrate differences between emulation and discovery tasks. First, human local understanding of task decision boundary $Y^H$ is collapsed with $Y$ in emulation tasks (Figure 3b), so no edge goes into $Y^H$, and $Y^H$ affects $Z^H$ and $\hat{Y}^H$. However, unlike in emulation tasks where $Y^H \equiv Y$, the edge connections remain the same in discovery tasks (Figure 3c), i.e., we are unable to rule out any connections for now. Hence, human understanding of task decision boundary is usually not of interest in emulation tasks (Chandrasekaran et al., 2018; Nguyen, 2018). In comparison, human understanding of both model decision boundary and task decision boundary has been explored in discovery tasks (Bansal et al., 2020; Feng & Boyd-Graber, 2018; Park et al., 2019; Wang & Yin, 2021).

---

[8]Emulation and discovery can be seen as two ends of a continuous spectrum. The emulation vs. discovery categorization determines the set of causal diagrams that applies to the problem; this decision is at the discretion of practitioners who design and interpret experiments using our framework.

*Condition 2—prediction shown vs. hidden (Effect of show($\hat{Y}$)).* An alternative condition is an intervention that shows the model prediction $\hat{Y}$ to the human. In this way, human would then gain a perfect understanding of the *local* model decision boundary and always predict $\hat{Y}^H = \hat{Y}$.

We start with emulation tasks, where the relationships are relatively straightforward because the human understanding of task decision boundary is perfect ($Y^H \equiv Y$). When $\hat{Y}$ is shown (Figure 3d), human understanding of local predicted label becomes perfect, i.e., $\hat{Y}^H \equiv \hat{Y}$. It follows that $Z^H = I(Y^H \neq \hat{Y}^H) = I(Y \neq \hat{Y}) = Z$. This scenario happens in debugging for emulation tasks, where model developers know the true label, the predicted label, and naturally whether the predicted label is incorrect for the given instance. In this case, the desired understanding is not local, but about global model decision boundary. Refer to Appendix D for discussions on global understanding. In comparison, when $\hat{Y}$ is not shown (e.g., an auditor tries to extrapolate the model prediction), recall $Y^H \equiv Y$ in emulation tasks, so $Y^H$ can affect $\hat{Y}^H$ and $Z^H$. As shown in Figure 3e, the connection between $\hat{Y}^H$ and $Z^H$ remains unclear.

In discovery tasks, when $\hat{Y}$ is shown (Figure 3f), $\hat{Y}^H \equiv \hat{Y}$. However, the relationships between $Y^H$ and $Z^H$ remain unclear and can be potentially shaped by further information such as machine explanations. When $\hat{Y}$ is not shown (Figure 3g), no information can be used to update the causal diagram in discovery tasks. Therefore, Figure 3g is the same as the base diagram where all interactions between local understandings are possible, which highlights the fact that no insights about human understandings can be derived without any assumption or intervention.

**Implications.** Our framework reveals the underlying mechanism of human local understanding with two important conditions: 1) knowing the task decision boundary; and 2) showing machine predictions $\hat{Y}$. Such conditions allow us to disambiguate connections between human understanding of core variables. For example, in emulation with prediction shown (Figure 3d), the relationship between all variables is simplified to a deterministic state.

Another implication is that we need to make explicit assumptions in order to make claims such as human performance improves because human understanding of the model error is better (i.e., humans place appropriate trust in model predictions). As there exist ambiguous links between variables, for example, in discovery tasks with prediction shown (Figure 3f), we can not tell whether it is $Y^H \rightarrow Z^H$ or $Z^H \rightarrow Y^H$, nor can we tell from experimental data without making assumptions. The alternative hypothesis to "appropriate trust $\rightarrow$ improved task performance" is that $\hat{Y}$ directly improves human understanding of the task decision boundary directly (i.e., "model predictions $\rightarrow$ improved task performance"). In these ambiguous cases, explanations may shape which scenario is more likely, and it is critical to make the assumptions explicit to support causal claims drawn from experiments.

## 5 Machine Explanations and Human Intuitions

In this section, we discuss the utility and limitations of machine explanations using our framework. We first show that without assumptions about human intuitions, explanations can improve human understanding of model decision boundary, but not task decision boundary or model error. As a result, *complementary performance in discovery tasks is impossible*. We then discuss possible ways that human intuitions can allow for effective use of explanations and lay out several directions for improving the effectiveness of explanations. Our analyses highlight the importance of articulating and measuring human intuitions in using machine explanations to improve human understanding.

### 5.1 Limitations of Machine Explanations without Human Intuitions

**Existing explanations are generated from $g$ (Figure 4(a)).** We first introduce explanation ($E$) to our causal diagram. Since the common goal of explanation in the existing literature is to explain the underlying mechanism of the model, $E$ is derived from a function, $e(g)$, which is independent from $f$ given $g$ For example, gradient-based methods use gradients from $g$ to generate explanations (Bach et al., 2015; Sundararajan et al., 2017). Both LIME (Ribeiro et al., 2016) and SHAP (Lundberg & Lee, 2017) use local surrogate models to compute importance scores, and the local surrogate model is based on $g$. Counterfactual

explanations (Mothilal et al., 2019; Wachter et al., 2017) typically identify examples that lead to a different predicted outcome from $g$. As a result, $e(g)$ and $E$ do not provide information about $f$ or $z$ beyond $g$.

In addition, we emphasize that there should be no connection between $E$ and task-specific intuitions, $H$. Conceptually, only task-agnostic human intuitions are incorporated by existing algorithms of generating explanations. It is well recognized that humans cannot understand all parameters in a complex model, so promoting sparsity incorporates some human intuition. Similarly, the underlying assumption for transparent models is that humans can fully comprehend a certain class of models, e.g., decision sets (Lakkaraju et al., 2016) or generalized linear models (Caruana et al., 2015; Nelder & Wedderburn, 1972). However, none of these assumptions on human intuitions are task-specifc, i.e., related to task decision boundary, model error, or human understanding of them.

Now we discuss the effect of explanations on human understanding without assuming any task-specific human intuitions (i.e., without adding new edges around $H$).

**Without assumptions about task-specific human intuitions, explanations can improve human understanding of model decision boundary, but cannot improve human understanding of task decision boundary or model error.** We start with the cases *where predicted labels are not shown.*

**Theorem 1.** *When predicted labels are not shown for an instance (X), explanation (E) that is derived from $g$ alone can improve human understanding of the model decision boundary (i.e., increasing the correlation between $\hat{Y}^H$ and $\hat{Y}$).*

*Proof.* Figure 4(b1) shows the subgraph related to $\hat{Y}$ and $\hat{Y}^H$ from Figure 2. Without explanations, $\hat{Y}$ and $\hat{Y}^H$ are independent given $X$. Figure 4(b2) demonstrates the utility of machine explanations. Because of the connection with $\hat{Y}$ through $g$ and $e(g)$, the introduction of $E$ *can* improve human understanding of model decision boundary, $\hat{Y}^H$. □

Note that our discussion on improvement is concerned with the upper bound of understanding assuming that humans can rationally process information if the information is available. This improvement holds regardless of the assumption about human knowledge of $Y$ (i.e., both in emulation and discovery tasks).

*When predicted labels are shown*, improving local human understanding of model decision boundary is irrelevant, so we focus on task decision boundary and model error.

**Theorem 2.** *When predicted labels are shown for an instance (X), explanation (E) that is derived from only $g$ cannot improve human understanding of task decision boundary or model error (i.e., increasing the correlation between $Y$ and $Y^H$ or the correlation between $Z$ and $Z^H$) beyond the improvement from showing predicted labels.*

*Proof.* First, in emulation tasks, when provided with predicted labels (i.e., $show(\hat{Y})$), humans would achieve perfect accuracy at approximating the three core variables. That is, the perfect local understanding already holds in emulation tasks without machine explanations, explanations have no practical utility in this setting. Therefore, machine explanations cannot help humans achieve better local approximation than showing predicted labels in local understanding.

In comparison, Figure 4(c1) shows the diagram for the more interesting case, discovery tasks. According to d-separation, given the prediction $g$ and $X$, the explanation $E$ is independent of $Y$ and $Z$. That is, $E$ cannot provide any extra information about $Y$ (task decision boundary) and $Z$ (model error) beyond the model ($g$). As our definition about $g$ and $\hat{Y}$ in Section 3 entails, the model cannot provide any better approximation of $Y$ than $\hat{Y}$.[9] For a person who has no intuitions about a task at all, the best course of action is to follow machine predictions. Therefore, if we do not make assumptions about human intuitions, $E$ cannot bring any additional utility over showing $\hat{Y}$.

□

---

[9]In other words, $g$ cannot provide a bad $\hat{Y}$ while offering useful information in $E$ that can be directly used to improve $\hat{Y}$ for $g$ itself.

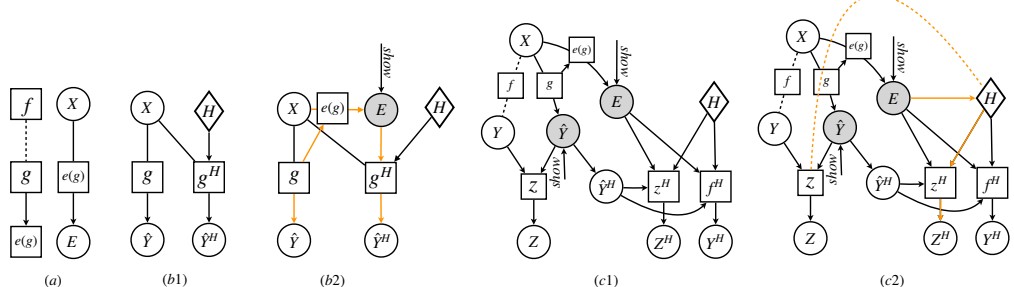

Figure 4: (a) $E$ is generated from $g$. (b1) $\hat{Y}$ and $\hat{Y}^H$ are independent given $X$. (b2) The utility of $E$: $E$ can improve human understanding of model decision boundary. (c1) $E$ cannot improve human understanding of task decision boundary and model error without human intuitions. (c2) Combined with human intuitions, $E$ can improve task decision boundary and model error. We use orange lines to highlight the links that lead to the utility of $E$.

An important corollary is thus

**Corollary 2.1.** *Complementary performance is impossible in discovery tasks without extra assumptions about human intuitions.*

Our results suggest that the common hypothesis that explanations improve human decision making, i.e., bringing $Y^H$ closer to $Y$, hinges on assumptions about $H$. Note that the scope of these two theorems is for local human understanding, please refer to Appendix D for a discussion on global understanding.

As a concrete example, consider the case of topic categorization with an alien who does not understand human language (i.e., guaranteeing that there is no intuition about the task). Machine explanation such as feature importance cannot provide meaningful information about model error to the decision maker (the alien). For instance, highlighting the word "basketball" as an important feature can help the alien understand the model decision boundary and learn that "basketball" is associated with the sports category from the model's perspective (Theorem 1). However, due to the lack of any intuition about human language, the alien interprets "basketball" as a string of bits and cannot achieve any improved understanding in what constitutes the sports category (i.e., the task decision boundary) and model error (Theorem 2). This extreme example illustrates how the lack of relevant task-specific intuitions may limit the utility of explanations in deceptive review detection (Lai & Tan, 2019). Our experiments in Section 6 will simulate a case without human intuitions.

In comparison, machine explanations have been shown to be useful in model debugging scenarios (Ribeiro et al., 2016). In this case, the implicit assumption about human intuition is that developers know the groundtruth (task decision boundary) and explanations derived from $g$ can improve the understanding of model decision boundary, which in turn improves the understanding of model error. This improved understanding of model error is essential to model debugging.

## 5.2 Machine Explanations + Human Intuitions

Next, we discuss how explanations can be integrated with human intuitions to achieve an improved understanding in discovery tasks (recall that $Z^H$ and $Y^H$ are entailed in emulation tasks when $\hat{Y}$ is shown). We have seen that $E$ itself does not reveal more information about $Y$ or $Z$ beyond $g$. Therefore, an important role of $E$ is in shaping human intuitions and influencing the connection between $f, g, z$ with $H$. We present two possible ways.

**Activating existing knowledge about model error.** $E$ can activate existing human knowledge that can reveal information about model error (Figure 4(c2)). We examine two sources of such knowledge that are concerned with *what* information should be used and *how*. First, human intuitions can evaluate *relevance*, i.e., whether the model leverages relevant information from the input based on the explanations. For example, human intuitions recognize that "Chicago" should not be used for detecting deceptive reviews or that race should not be used for recidivism prediction, so a model prediction relying on these signals may be more

likely wrong. The manifestation of relevance depends on the explanation's form: feature importance directly operates on pre-defined features (e.g., highlighting race for tabular data or a word in a review), example-based or counterfactual explanations narrow the focus of attention to a smaller (relevant) area of the input.

Second, human intuitions can evaluate *mechanism*, i.e., whether the relationship between the input and the output is valid. Linear relationship is a simple type of such relation: human intuitions can decide that education is negatively correlated with recidivism, and thus that a model making positive predictions based on education is wrong. In general, mechanisms can refer to much more complicated (non-linear) relations between (intermediate) inputs and labels.

Figure 4(c2) illustrates such activations in causal diagrams. The link from $E$ to $H$ highlights the fact that human intuitions when $E$ is shown are different from $H$ without $E$ because these intuitions about relevance and mechanism would not have been useful without machine explanation. We refer to $H$ in Figure 4(c2) as $H_{show(E)}$. If $H_{show(E)}$ is correlated with $z$ (indicated by the dash link), then $Z^H$ is no longer independent from $Z$ (e.g., incorrect connection between education and recidivism is correlated with model error) and can thus improve $Z^H$ and then $Y^H$ because $Z$ is a collider for $Y$ and $\hat{Y}$, leading to complementary performance. It is important to emphasize that this potential improvement depends on the quality of $H_{show(E)}$. Note that in this case, the improvement in $Z^H$ also disambiguates the relationship between $Y^H$ and $Z^H$ (i.e., appropriate trust → improved decision making). **The lack of useful task-specific human intuitions can explain the limited human-AI performance in deceptive review detection (Lai & Tan, 2019).**

**Expanding human intuitions.**  Another way that explanations can improve human understanding is by expanding human intuitions (see Figure 10 in the Appendix). As an example of counterintuitive explanations, the word "Chicago" is highlighted as an important factor for deceptive review (Lai et al., 2020) for two reasons: 1) deceptive reviews in this dataset are written by crowdworkers for hotels in Chicago; 2) people are less likely to provide specific details (such as which street/specific neighborhood) when they write fictional texts. Highlighting the word for "Chicago" (relevance) and its connection with deceptive reviews (mechanism) is counterintuitive to humans because this is not part of existing human knowledge. But if machine explanations can expand human intuitions and help humans derive theory I, this can lead to improvement in human understanding of task decision boundary (i.e., humans develop new knowledge from machine explanations). Formally, the key change in the diagram for this scenario is that $E$ influences human intuitions in the next time step $H_{t+1}$.

### 5.3   Towards Effective Explanations for Improving Human Understanding

Machine explanations are only effective if we take into account human intuitions. We encourage the research community to advance our understanding of task-specific human intuitions, which are necessary for effective human-AI decision making. We propose the following recommendations.

**Articulating and measuring human intuitions.**  It is important to think about how machine explanations can be tailored to either leveraging existing human knowledge or expanding human intuitions, or other ways that human intuitions can work together with explanations.

First, we need to make these assumptions about human intuitions explicit so that the research community can collectively study them rather than repeating trial-and-error with the effect of explanations on an end outcome such as task accuracy. We recommend the research community be precise about the type of tasks, the desired understanding, and the required human intuitions to achieve success with machine explanations.

Second, to make progress in experimental studies with machine explanations, we need to develop ways to either control or measure human intuitions. This can be very challenging in practice. We will present an experiment where we control and measure human intuitions in human-AI decision making as an initial step.

**Incorporate $f$ and $z$ into explanations.**  An important premise for explanations working together with human intuitions is that machine explanations capture the mechanism or the relevance underlying the model. Indeed, faithfulness receives significant interest from the machine learning community for the sake of

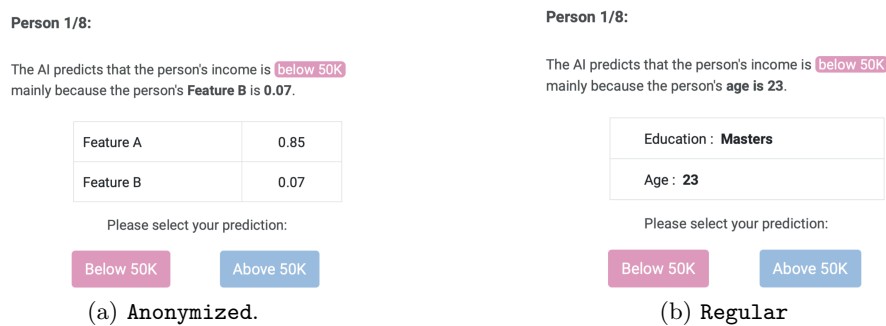

Figure 5: Screenshots of the interfaces for the anonymized group (a) and the regular group (b).

explaining the mechanisms of a model. However, faithfulness to $g$ alone is insufficient to improve human understanding of task decision boundary and the model error.

In order to effectively improve human understanding of $f$ and $z$, it would be useful to explicitly incorporate $f$ and $z$ into the generation process of $E$. For example, a basic way to incorporate $z$ is to report the error rate in a development set. In the case of deceptive review detection, it could be when "Chicago" is used as an important feature, the model is 90% accurate. This allows humans to have access to part of model error and have a more accurate $Z^H$.

To summarize, we emphasize the following three takeaways:

- Current machine explanations are mainly about the model and its utility for human understanding of the task decision boundary and the model error is thus limited.

- Human intuitions are a critical component to realize the promise of machine explanations in improving human understanding and achieving complementary performance.

- We need to articulate our assumptions about human intuitions and measure human intuitions, and incorporate human intuitions, $f$, and $z$ into generating machine explanations.

# 6  Experiments

In light of our formal framework, we emphasize the importance of incorporating human intuition for effective explanations. We designed our experiment as an example of applying our theoretical framework, to demonstrate how our framework can inform experimental design and analysis. While our experiment doesn't directly validate the theorems, it provides strong supporting evidence for the converse, illustrating that by explicitly articulating assumptions about human intuition, we can begin to predict situations and mechanisms in which explanations can be beneficial.

Our experiments test two hypotheses about the impact of human intuition on their interaction with machine explanations. First, when people do not have sufficient intuition to judge whether the model is correct, they are more likely to agree with the model (**H1**; see discussion on Figure 4(c1)). Second, when people do have intuition, they are more likely to agree with the model when model explanations are consistent with their intuitions (**H2**; see discussion on Figure 4(c2)).

To simulate real-world decision making, we follow the standard cooperative setting where the participants make predictions with the assistance of both model prediction and feature importance explanation. We use a synthetic model that allows us to create feature importance explanations such that either the participant has no intuition about the highlighted feature or the highlighted feature importance agrees/disagrees with the participant's intuition.

## 6.1  Experiment Design

We focus on the key considerations here and more details can be found in Appendix F.

**Task: income prediction.** Our hypotheses are about how people's decisions are affected by the alignment between their intuitions and the information presented about the model. So it's crucial that we can manipulate this alignment and control for confounders. Inspired by the Adult Income dataset (Blake, 1998), we choose the task of predicting a person's annual income based on their profile because people generally have intuitions about what factors determine income but are unlikely to know every person's income (hence a discovery task). We simplify the available features so that we can control for confounders. In our version, each profile contains two attributes, age and education, and the participant makes a binary prediction: whether the person's annual income is above or below $50K.

**Human intuition in income prediction.** Following Section 5.2, we consider human intuitions on relevance and mechanism. We define relevance ($\mathbf{R}$) as the intuition that education is more important than age in determining income, and mechanism ($\mathbf{M}$) as the intuition that income positively correlates with both features.

To better understand what these intuitions entail, let's consider Alice who believes $\mathbf{R}$ and $\mathbf{M}$. For a profile with high education and low age, as Alice believes that education is more important than age ($\mathbf{R}$), she will rely more on education; as she believes that education positively correlates with income, she will likely predict high income without any AI assistance. In Figure 6, We use the background color to represent Alice's likely predictions: blue for high income and red for low income. When machine prediction and explanation are shown, these intuitions can be used to make predictions.

**Alignment with human intuition.** Next, we explain how we implement explanation alignment, still using participant Alice as the example. Intuitively, alignment with $\mathbf{R}$ is determined by whether the explanation highlights the right feature. Since $\mathbf{R}$ specifies that education is more important than age, a feature importance explanation that's aligned with $\mathbf{R}$ should highlight education. Similarly, alignment with $\mathbf{M}$ is determined by whether the model prediction is consistent with the highlighted feature: if the highlighted feature has a high value, the model should predict high income. In Figure 6, we use a cross to visualize the violation of $\mathbf{R}$ and border for the violation of $\mathbf{M}$. Note that even if the explanation violates $\mathbf{R}$ (i.e., highlighting age instead of education), $\mathbf{M}$ can be supported if high age $\rightarrow$ high income or low age $\rightarrow$ low income.

**Measuring human intuitions.** Participants may not necessarily hold these assumed intuitions (i.e., $\mathbf{R}$ and $\mathbf{M}$), we thus measure human intuitions by asking participants about which feature is more important and whether the correlation is positive (see Appendix Section F.3).

**Removing human intuition.** To study H1, we need to "remove" human intuitions. To do that, we anonymize the features (education and age) to feature A and feature B. Figure 5a provides an example profile with removed intuitions. As a result, participants should have limited intuitions about this task, especially on relevance.

**Synthetic data.** To understand human decision making with the information provided from the model, we consider four groups of data in the table in Figure 6: AB are consistent with both $\mathbf{R}$ and $\mathbf{M}$, CD violates $\mathbf{R}$, EF violates $\mathbf{M}$, and GH violates $\mathbf{R}$ and $\mathbf{M}$. Figure 6 also presents a pictorial view. A-H enumerates all possible combinations of logically consistent explanations and predictions in the two off-diagonal quadrants. We focus on these two quadrants because 1) relevance does not matter for profiles that are high education & high age or low education & low age, and 2) participants are more likely to ignore information from our models in the other two quadrants.

**Agreement as the evaluation metric.** Since our hypotheses are formed around the agreement between human and model predictions, the focus is on model error. We infer $Z^H$ by measuring the agreement between human prediction with AI assistance ($Y^H$) and model prediction ($\hat{Y}$).

**Summary.** To summarize, there are two groups in our experiment, `regular` and `anonymized`. Each participant sees eight (ABCDEFGH) randomly shuffled examples. Since we evaluate two intuitions, we extend our two main hypotheses into the following four. We use agreement($\cdot$) to represent the average agreement on multiple data groups.

| R | M | $\hat{Y}$ | Identifier | Education | Age | Prediction | Explanation |
|---|---|---|---|---|---|---|---|
| ✓ | ✓ | ✓ | 🔵 A | High | Low | >50K | Education |
| | | | 🔴 B | Low | High | <50K | Education |
| ✗ | ✓ | ✗ | ⊗ C | High | Low | <50K | Age |
| | | | ⊗ D | Low | High | >50K | Age |
| ✓ | ✗ | ✗ | 🔵 E | High | Low | <50K | Education |
| | | | 🔴 F | Low | High | >50K | Education |
| ✗ | ✗ | ✓ | ⊗ G | High | Low | >50K | Age |
| | | | ⊗ H | Low | High | <50K | Age |

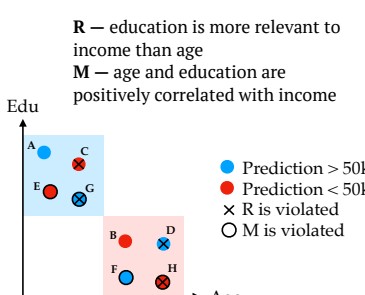

R — education is more relevant to income than age
M — age and education are positively correlated with income

🔵 Prediction > 50k
🔴 Prediction < 50k
✕ R is violated
◯ M is violated

Note: In columns R and M, ✓(✗) means the type of human intuition is supported (violated). In column $\hat{Y}$, ✓(✗) means the label given by Alice's intuition is the same as (different from) the predicted label.

Figure 6: We present our instances on the left and visualize them on the right. The background color represents Alice's likely predictions: blue (red) means $> \$50K$ ($< \$50K$).

- **H1** (Over-reliance without intuition): Without sufficient intuition, people tend to follow model predictions: agreement(`anonymized`) > agreement(`regular`).

- **H2a** (Alignment with **R** & **M** increases agreement): In the `regular` group, people are more likely to agree with model predictions when their intuitions are supported: agreement(AB) > agreement(GH). This hypothesis is an ideal test on "explanation → improved understanding of model error" because model predictions are the same in GH as AB, any observed difference should be attributed to the effect of explanations.

- **H2b** (Alignment with **R** correlates with agreement): In the `regular` group, people are more likely to agree with model predictions when **R** is supported: agreement(AB) > agreement(CD).

- **H2c** (Alignment with **M** correlates with agreement): In the `regular` group, people are more likely to agree with model predictions when **M** is supported: agreement(AB) > agreement(EF).

For **H1**, we use $t$-test to compare subjects in different conditions. For **H2**, we have a within-subject design and use paired $t$-test to compare the agreement rate of the same participants for instances with different identifiers.

**Limited validity of H2b, H2c for evaluating explanation consistency.** We emphasize that although H2b and H2c are reasonable hypotheses, they cannot support the causal effect of explanation consistency. As shown in Figure 3(f), the link between $Y^H$ and $Z^H$ is unclear in discovery tasks when predicted labels are shown. With our simplified setup, when only **R** or **M** is violated, $Y^H$ is already different from $\hat{Y}$ without any assistance. Even if explanation does not shape $H$ (Figure 4(c1)) (i.e., humans ignore machine explanations), H2b and H2c can hold. In contrast, H2a can only hold when explanation works together with human intuition and $Z^H \to Y^H$ (Figure 4(c2)). In other words, as predictions are the same in GH as AB, any observed difference should be attributed to the effect of explanations. This discussion showcases the utility of our framework in experiment design.

## 6.2 Study Details

We use crowdsourcing platform Prolific[10] and recruit 242 participants; 136 in the `regular` group, 106 in the `anonymized` group, following our power analysis. Participants' age range from 18 to 79 with an average of 38. There are 121 female and 121 male participants. Participation was limited to adults in the US and first language is English. We also prescreen participants based on their approval rate (97% to 100%). The median time taken for each participant is 3.32 minutes and they are paid $0.60, which leads to an hourly payment of $10.85. Among the `regular`, 70 hold the assumed intuitions.

Participants are first presented with brief information about the study and a consent form. Next, for the `regular` group, to measure their intuitions, they are asked about which feature is more important and

---

[10]https://prolific.co.

whether the correlation is positive. For `anonymized` group, they skip this step. Then, participants proceed to complete the main part of the study, in which they answer 8 adult income prediction questions. Figure 5 shows screenshots of the question answering page for the two groups. The order of instances is randomized across participants. As a final step, they complete an exit survey.

**Data.** Specifically, we construct 'high education low age' samples as masters with age varying from 23 to 26. 'low education high age' samples are middle school with age varying from 46 to 49. To avoid the task being too artificial, we use different ages for A-H for a participant. To further remove the potential effect due to the differences between ages, we create two groups of data so that the average age is the same for ABCD and EFGH. The full dataset can be found in Appendix Table 3. We randomly assign participants to a data group and ensure that each group is assigned to a similar number of participants.

### 6.3 Results

**H1 (Over-reliance without intuition).** In order to compare the agreement across two conditions (anonymized vs. regular group), we compute user-level agreement in each condition and run the independent $t$-test between users in the two conditions. Consistent with H1, our results show that users in the `anonymized` group are more likely to agree with AI compared with users in the `regular` group ($t = 7.29, p < 0.001$). The average agreement rate for users in the `anonymized` group is as high as 70.64%, compared to 54.32% in the `regular` group. In other words, without any strong intuitions about the underlying task, humans tend to rely on AI predictions.

**H2a (Alignment with R & M; explanation consistency).** Consistent with the hypothesis, the average agreement rate of AB is 90.71% and GF is 83.57%, and the difference is statistically significant ($t = 2.44, p = 0.017$). This result is consistent with early work on explanation coherence (Thagard, 1989).

**H2b (Alignment of R; explanation consistency).** We use paired $t$-test on the users that hold the assumed intuitions in Figure 6. We first investigate the agreement of **R**. Consistent with our hypothesis, the agreement of AB is 90.71%, much higher than CD (25.00%). The difference is statistically significant ($t = 14.25, p < 0.001$).

**H2c (Alignment with M; explanation consistency).** Similarly, we investigate agreement for alignment with **M**. Consistent with our hypothesis, the agreement with AB is 90.71%, much higher than that with EF (22.14%). The difference is statistically significant ($t = 15.01, p < 0.001$).

In summary, results from our experiment have confirmed all the hypotheses, and demonstrate how task-specific human intuitions shape the outcome of a study on the effect of machine explanations.

## 7 Conclusion

In this work, we propose the first theoretical work to formally characterize the interplay between machine explanations and human understanding. We theoretically prove the necessity of incorporating human intuition for effective explanations. Effective human-AI decision making will require formalization of both human and AI's contribution in decision making. To this end, we recommend the research community explicitly articulate human intuitions involved in research hypotheses and experimental design. Hypotheses such as "explanations improve human decisions" can only provide limited insights because they are dependent on specific human intuitions and may not be generalizable to other contexts. In algorithm development, in order to effectively improve human understanding of $f$ and $z$, it would be useful to explicitly incorporate $f$ and $z$ into generating machine explanations.

## Acknowledgement

We thank the anonymous reviewers for their insightful comments. We also thank Victor Veitch and Vera Q. Liao for their insightful comments and suggestions, Giang Nguyen for suggesting relevant citations, and

members of the Chicago Human+AI lab for their thoughtful feedback. This paper is supported by in part by a CDAC discovery grant at the University of Chicago and an NSF grant, IIS-2040989.

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

## A   Limitations and Potential Negative Societal Impacts

**Limitations.**   We present a deterministic example for the three core concepts in the paper for ease of understanding. In general, one can also think of model decision boundary, task decision boundary, and model error probabilistically. Our three core concepts can be applied in most classificaiton tasks. However, for other tasks like generation, the notion of model decision boundary and model error require substantial future work.

Our theoretical framework is only a first step towards understanding the interplay between machine explanations and human understanding. For instance, we do not consider the effect of showing $\hat{Y}$ on human intuitions. Our discussions are mostly based on information entailed by causal diagrams without accounting for psychological biases in human intuitions (e.g., issues related to numeracy (Lai & Tan, 2019; Peters et al., 2006)). Our work focuses on understanding the role of machine explanations, while another recent study provides a deep dive related to our discussion in Section 4 by examining human-AI collaboration in different distributions (regimes) of instances (Donahue et al., 2022).

Our work addesses the quantitative measures of human understanding in the context of human-AI decision making in classificaiton tasks. More generally, Hoffman et al. (2018) aims to summarize a variety of psychometric measurements for explainable AI.

**Potential negative societal impacts.**   Our work is highly theoretical, and can shed light on both positive and negative societal impacts of interpretable machine learning. The potential negative impacts mainly from the possible downstream applications. The implications drawn from our work that researchers should incorporate human intuitions into explanation methods can affect many AI applications in the real-world. The risks arise especially in high-stake domains, such as recidivism task and healthcare. Black box machine learning models have shown bias in a lot of applications. The way that human intuition should be included in generating explanations can further pose AI bias into human decision making process if used inappropriately, thus exacerbating existing inequalities. For example, when human intuition are exploited in a way that the human decision maker will more likely to follow AI predictions regardless, any existing algorithmic bias in this case may exacerbates social inequities.

## B   A summary of recent empirical studies

We surveyed literature of recent empirical studies that include quantifiable metric for evaluating human understanding with machine explanations. In Table 2 we present a summary of recent empirical studies and provide a reinterpretation with the three core concepts: model decision boundary $g$, model error $z$, and task decision boundary $f$, as we defined in the main paper.

## C   Notation Clarification & Gate notation for the *show* operator

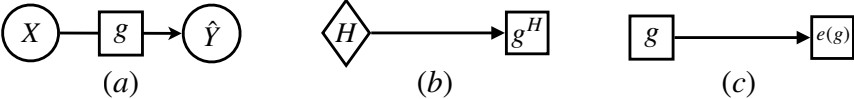

Figure 7: Adapted causal diagrams.

We provide additional clarification and justification for our adapted causal diagrams.

- Figure 7(a): $g$ captures the relation between $X$ and $\hat{Y}$. This is implicit in standard causal diagrams. However, given that our discussion centers around human understanding of these relations, we make them explicit and annotate them on edges.

- Figure 7(b): a new type of node $H$ is indicated by the diamond shape. $H$ captures human task-specific intuitions. It generates human approximations of the core concepts. As a result, it has directed links to functions in our diagrams. Our goal in this work is to provide a formal language to discuss human

| Paper | Model | Prediction | Explanations | g | z | f |
|---|---|---|---|---|---|---|
| Colin et al. (2022) | InceptionV1, ResNet | Hidden | Local feature importance (Saliency,Gradient Input, Integrated Gradients, Occlusion (OC), SmoothGrad (SG) and Grad-CAM) | ✓ | ✗ | ✗ |
| Taesiri et al. (2022) | ResNet, kNN, other deep learning models | Shown | Confidence score, example-based methods (nearest neighbors) | ✗ | ✓ | ✓ |
| Kim et al. (2022) | CNN, BagNet, ProtoPNet, ProtoTree | Mixed | Example-based methods (ProtoPNet, ProtoTree), local feature importance (GradCAM, BagNet) | ✓ | ✓ | ✓ |
| Nguyen et al. (2021) | ResNet | Shown | Model uncertainty (classification confidence (or probability)); Local feature importance (gradient-based, salient-object detection model); Example-based methods (prototypes) | ✗ | ✓ | ✓ |
| Buçinca et al. (2021) | Wizard of Oz | Shown | Model uncertainty (classification confidence (or probability)) | ✗ | ✓ | ✓ |
| Chromik et al. (2021) | Decision trees/random forests | Shown | Local feature importance (perturbation-based SHAP) | ✓ | ✗ | ✗ |
| Nourani et al. (2021) | Other deep learning models | Shown | Local feature importance (video features) | ✓ | ✓ | ✓ |
| Liu et al. (2021) | Support-vector machines (SVMs) | Shown | Local feature importance (coefficients) | ✓ | ✓ | ✓ |
| Wang & Yin (2021) | Logistic regression | Shown | Example-based methods (Nearest neighbor or similar training instances); Counterfactual explanations (counterfactual examples); Global feature importance (permutation-based); | ✓ | ✓ | ✗ |
| Poursabzi-Sangdeh et al. (2021) | Linear regression | Shown | Presentation of simple models (linear regression); Information about training data (input features or information the model considers) | ✓ | ✓ | ✓ |
| Bansal et al. (2020) | RoBERTa; Generalized additive models | Shown | Model uncertainty (classification confidence (or probability)); Local feature importance (perturbation-based (LIME)); Natural language explanations (expert-generated rationales); | ✗ | ✓ | ✓ |
| Zhang et al. (2020) | Decision trees/random forests | Shown | Model uncertainty (classification confidence (or probability)); Local feature importance (perturbation-based SHAP); Information about training data (input features or information the model considers) | ✗ | ✓ | ✓ |
| Abdul et al. (2020) | Generalized additive models | Shown | Global feature importance (shape function of GAMs) | ✓ | ✗ | ✗ |
| Lucic et al. (2020) | Decision trees/random forests | Hidden | Counterfactual explanations (contrastive or sensitive features) | ✓ | ✗ | ✗ |
| Lai et al. (2020) | BERT; Support-vector machines | Shown | Local feature importance (attention); Model performance (accuracy); Global example-based explanations (model tutorial) | ✗ | ✓ | ✓ |
| Alqaraawi et al. (2020) | Convolution Neural Networks | Hidden | Local feature importance (propagation-based (LRP), perturbation-based (LIME)) | ✓ | ✗ | ✗ |
| Carton et al. (2020) | Recurrent Neural Networks | Shown | Local feature importance (attention) | ✗ | ✓ | ✓ |
| Hase & Bansal (2020) | Other deep learning models | Shown | Local feature importance (perturbation-based (LIME)); Rule-based explanations (anchors); Example-based methods (Nearest neighbor or similar training instances); Partial decision boundary (traversing the latent space around a data input) | ✓ | ✗ | ✗ |
| Buçinca et al. (2020) | Wizard of Oz | Mixed | Example-based methods (Nearest neighbor or similar training instances) | ✓ | ✓ | ✓ |
| Kiani et al. (2020) | Other deep learning models | Shown | Model uncertainty (classification confidence (or probability)); Local feature importance (gradient-based) | ✗ | ✓ | ✓ |
| Gonzalez et al. (2020) | Other deep learning models | Shown | extractive evidence | ✗ | ✓ | ✓ |
| Lage et al. (2019) | Wizard of Oz | Mixed | Rule-based explanations (decision sets) | ✗ | ✓ | ✓ |
| Weerts et al. (2019) | Decision trees/random forests | Hidden | Model uncertainty (classification confidence (or probability)); Local feature importance (perturbation-based SHAP) | ✗ | ✓ | ✓ |
| Guo et al. (2019) | Recurrent Neural Networks | Shown | Model uncertainty (classification confidence (or probability)) | ✗ | ✗ | ✓ |
| Friedler et al. (2019) | Logistic regression; Decision trees/random forests; Shallow (1- to 2-layer) neural networks | Hidden | Counterfactual explanations (counterfactual examples); Presentation of simple models (decision trees, logistic regression, one-layer MLP) | ✓ | ✗ | ✗ |
| Lai & Tan (2019) | Support-vector machines | Shown | Example-based methods (Nearest neighbor or similar training instances); Model performance (accuracy) | ✗ | ✓ | ✓ |
| Feng & Boyd-Graber (2018) | Generalized additive models | Shown | Model uncertainty (classification confidence (or probability)); Global example-based explanations (prototypes) | ✗ | ✓ | ✓ |
| Nguyen (2018) | Logistic regression; Shallow (1- to 2-layer) neural networks | Hidden | Local feature importance (gradient-based, perturbation-based (LIME)) | ✓ | ✗ | ✗ |
| Ribeiro et al. (2018) | VQA model (hybrid LSTM and CNN) | Hidden | Rule-based explanations (anchors) | ✓ | ✗ | ✗ |
| Chandrasekaran et al. (2018) | Convolution Neural Networks | Hidden | Local feature importance (attention, gradient-based) | ✓ | ✗ | ✗ |
| Biran & McKeown (2017) | Logistic regression | Shown | Natural language explanations (model-generated rationales) | ✗ | ✓ | ✓ |
| Lakkaraju et al. (2016) | Bayesian decision lists | Hidden | Rule-based explanations (decision sets) | ✓ | ✗ | ✗ |
| Ribeiro et al. (2016) | Support-vector machines; Inception neural network | Shown | Local feature importance (perturbation-based (LIME)) | ✓ | ✗ | ✗ |
| Bussone et al. (2015) | Wizard of Oz | Shown | Model uncertainty (classification confidence (or probability)) | ✗ | ✓ | ✗ |
| Lim et al. (2009) | Decision trees/random forests | Shown | Rule-based explanations (tree-based explanation); Counterfactual explanations (counterfactual examples) | ✓ | ✗ | ✗ |

Table 2: A summary of recent empirical studies measuring human understanding with machine explanations. The papers are sorted by time, starting from the newest. Note: columns $g$, $z$, and $f$ mean model decision boundary, model error, and task decision boundary respectively. ✓ (or ✗) means the study measures (or does not measure) the corresponding type of human understanding. This table is also maintained online at https://github.com/Chacha-Chen/Explanations-Human-Studies.

mental models in the context of human-AI decision making and articulate assumptions about $H$ in different conditions.

- Figure 7(c): We use relation between functions to indicate that explanations are generated with a function $(e(g))$ derived from $g$. Although the relations are between functions, they act in the same way as causal relations between variables, so d-separation still applies in the function space.

Our *show* operator can be rewritten using gate notation (Winn, 2012). For example, Figure 8 shows the diagram under the emulation condition; this diagram is equivalent to Figure 3(b) which uses our *show* notation. Since the gate is just an intervention written differently, a *show* operation can also be rewritten as a regular intervention. As a corollary, *show* does not represent conditional probability: in $\mathbb{E}[Y^H = Y | show(\hat{Y})]$ and $\mathbb{E}[Y^H = Y]$, $Y^H$ is generated by two distinct processes.

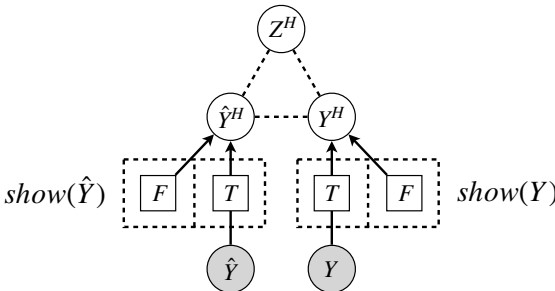

Figure 8: The partial realization of the base diagram under the emulation condition, visualized using gate notation; the semantics of the diagram is equivalent to Figure 3.b which uses our *show* operation.

# D    Global Understanding

Similar to Section 4, we discuss how interventions and assumptions can shape human global understanding. We discuss the two representative settings with concrete examples in (Figure 9) and the other settings are similar to derive.

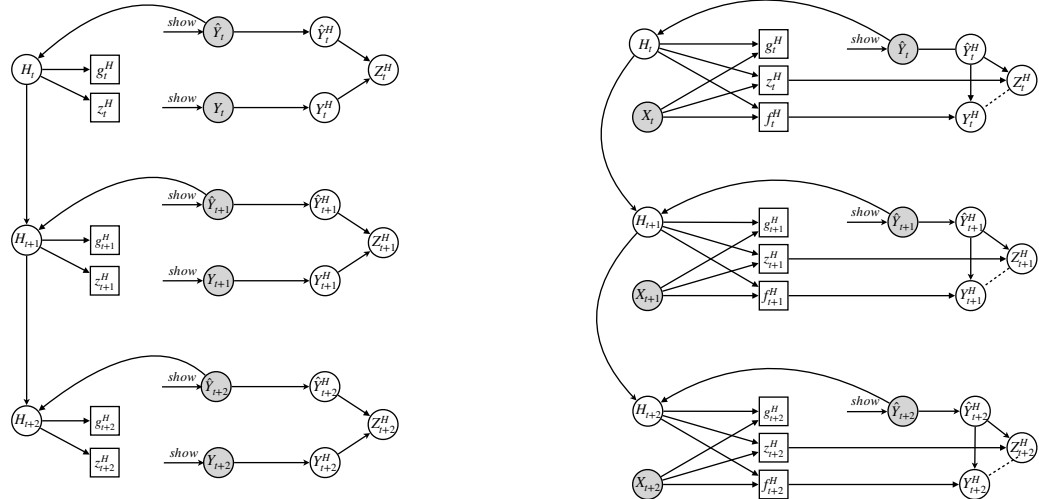

(*a*) Emulation: prediction shown (e.g. model debugging)          (*b*) Discovery: prediction shown (e.g. AI assisted decision making in recidivism task)

Figure 9: (a): Emulation task with prediction shown; (b): Discovery task with prediction shown. We use the dotted arrow to represent the temporal updates of human global understanding. In each of the setting, we illustrate 3 time steps from $t$ to $t + 2$.

**Emulation tasks with machine prediction shown (Figure 9(a)).** We start with a simple example, emulation tasks with machine prediction shown. Model debugging for developers is one such example. In this case, human intuition at time step $t$ derives human global understanding $g^H$ and $z^H$ directly. Since it is an emulation task, $f^H$ is not of interest, we omit it in the graph. For human local understanding, as $\hat{Y}$ is shown, $\hat{Y}^H \equiv \hat{Y}$. Also, given the emulation assumption, $Y^H \equiv Y$. We add subscript $t$ to indicate time steps.

This scenario captures the dynamics of debugging. At the human local understanding level, Figure 9(a) at each time step looks the same as Figure 3(d). As we have discussed, the relationship between local variables is deterministic. While knowing whether the model is correct for a single case may not be particularly useful for debugging, the key difference between the local version (Figure 3(d)) and the global version (Figure 9(a)) is the edge from observations ($\hat{Y}_t$) to human intuition $H_t$ and the temporal updates of human intuitions (the edge from $H_t$ to $H_{t+1}$). This allows model developers to update their belief of the model error $z^H$ and the decision boundary $g^H$ and decide the scope of this bug and whether this is a bug that needs to be fixed or can be fixed. The case of debugging highlights that global understandings are often the targets in human-AI interaction and updates in human belief is critical for understanding changes in global understanding.

For example, in Ribeiro et al. (2016), in topic classification, an emulation task where users know task decision boundary, showing LIME explanations can improve human global understanding of model decision boundary (i.e., which features are important) and thus also model error.

Explanation ($E$) can be incorporated in this figure and increase the connection between $g^H$ and $g$ because $g$ is a parent of $E$.

**Discovery task with prediction shown (Figure 9b).** A concrete example of this scenario is AI-assisted decision making in the recidivism task (Kleinberg et al., 2018), where judges make bailing decisions giving risk estimation ($\hat{Y}$). The update in $f^H$ indicates the fact that we are interested in improving global human understanding of task decision boundary. Similar to the local case, we do not know the exact relationship between $Z^H$ and $Y^H$ for the local variable. Therefore, an alternative path to improve $Y^H$ is by improving global human understanding of model error ($z^H$), then $Z^H$, leading to $Y^H$ with $\hat{Y}$ shown.

Similar to our discussion in Section 5, by showing machine predictions, human global understanding of $g^H$ can be updated. However, it is impossible to see updates on $z^H$ and $f^H$ beyond $g$ since human understanding is bounded by the ML model without assumptions about human intuitions. Combined with human intuitions and machine explanations $E$, $z^H$ and $f^H$ can be potentially updated.

**Implications.** The first takeaway is that improving global understanding is often the actual target of interventions. For instance, in emulation tasks with machine predictions shown, although local understandings have deterministic relations with each other, the key open question lies in how understanding local error relates to updates in the global understanding of model error.

Second, in order to improve global understanding, it is critical to understand the updates in human global understanding. Interventions are ideally designed so that $f^H \sim f$, $g^H \sim g$, $z^H \sim u$, depending on the application. Therefore, we observe that showing predicted labels only reveals information about $g$, and the information related to $f$ and $u$ is constrained by $g$. This observation highlights the limited utility of showing predicted labels.

Finally, our framework shows that there does not have to be links between $f^H$, $g^H$, and $z^H$. Although we may often be interested in conclusions such as calibrating trust improves human performance, the causal link may be unidentifiable and this relationship can also change over time. This observation reveals a fundamental limit in understanding the effect of machine explanations on human understanding. That said, it is possible to identify a causal relation between $Z^H$ and $Y^H$ by making additional assumptions on the diagrams (e.g., there is no temporal updates in human mental models, in other words, all time steps are independent of each other).

# E    Expanding Human Intuitions

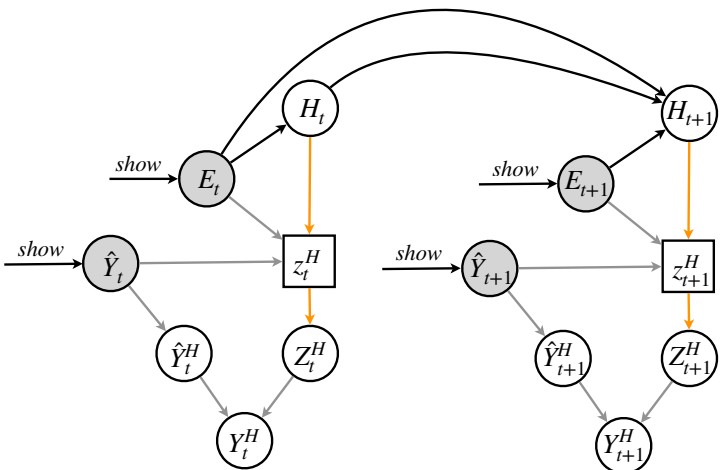

Figure 10: Explanations $E$ help expanding human intuitions by influencing human intuitions in the next time step.

# F    Additional Experiment Details

### F.1    Data

Table 3 shows the data instances we constructed and used in our experiment. Specifically, we construct 'high education low age' samples as masters with age varying from 23 to 26. 'low education high age' samples are middle school with age varying from 46 to 49. To avoid the task being too artificial, we use different ages for A-H for a participant. To further remove the potential effect due to the differences between ages, we create two groups of data so that the average age is the same for ABCD and EFGH. We randomly assign participants to a data group and ensure that each group is assigned to a similar number of participants.

| Instance ID | Education | Age | Feature A | Feature B | Predicted label (? \$50K) | Explanation | Identifier |
|---|---|---|---|---|---|---|---|
| 1 | Masters | 25 | 0.85 | 0.13 | > | Education | 🔵 A1 |
| 2 | Masters | 24 | 0.85 | 0.10 | > | Education | 🔵 A2 |
| 3 | Middle school | 46 | 0.15 | 0.83 | < | Education | 🔴 B1 |
| 4 | Middle school | 49 | 0.15 | 0.93 | < | Education | 🔴 B2 |
| 5 | Masters | 26 | 0.85 | 0.17 | < | Age | ⊗ C1 |
| 6 | Masters | 23 | 0.85 | 0.07 | < | Age | ⊗ C2 |
| 7 | Middle school | 48 | 0.15 | 0.90 | > | Age | ⊗ D1 |
| 8 | Middle school | 47 | 0.15 | 0.87 | > | Age | ⊗ D2 |
| 9 | Masters | 23 | 0.85 | 0.07 | < | Education | 🔴 E1 |
| 10 | Masters | 26 | 0.85 | 0.17 | < | Education | 🔴 E2 |
| 11 | Middle school | 47 | 0.15 | 0.87 | > | Education | 🔵 F1 |
| 12 | Middle school | 48 | 0.15 | 0.90 | > | Education | 🔵 F2 |
| 13 | Masters | 24 | 0.85 | 0.10 | > | Age | ⊗ G1 |
| 14 | Masters | 25 | 0.85 | 0.13 | > | Age | ⊗ G2 |
| 15 | Middle school | 49 | 0.15 | 0.93 | < | Age | ⊗ H1 |
| 16 | Middle school | 46 | 0.15 | 0.83 | < | Age | ⊗ H2 |

Table 3: The data instances used in our experiment.

### F.2 Participants' Agreement Report

In table 4, we report participants' average agreement across different groups and different categories (identifiers) of instances. We average over participants. Each participant did one question from each data instance category (identifier).

| Identifier | Agreement (`anonymized`) | Agreement (`regular`) |
|:---|:---:|:---:|
| ● A | 40.75% | 85.71% |
| ● B | 45.26% | 95.71% |
| ⊗ C | 41.41% | 32.86% |
| ⊗ D | 39.31% | 17.14% |
| ● E | 47.25% | 28.57% |
| ● F | 48.70% | 15.71% |
| ⊗ G | 48.45% | 74.29% |
| ⊗ H | 48.70% | 92.86% |

Table 4: Participants' agreement by the identifier in `anonymized` group

### F.3 User interface design

**Task description and attention check.** After participants agree with the online consent form (Figure 11), they will be re-directed to a page including a description of the task and an attention check, as shown in Figure 12 and Figure 13.

**Measuring human intuitions.** For `regular` group, participants are re-directed to the preliminary questionnaire page after they read the task description and passed the attention check. We calibrate human intuition in terms of relevance (R) and mechanism (M) using the 4 questions as shown in Figure 14.

**Exit survey.** After participants have done with all the prediction problems, they will proceed to an exit survey page, as shown in Figure 15.

**Please read the consent form below carefully before proceeding.**

To begin the study, please click "Agree" at the bottom and you will proceed to the detailed task instructions. At the end of the study, you will be redirected back to the Prolific website to validate that you have completed the study.

**Consent Form**

**Description**: We are researchers at the ⬚⬚⬚⬚⬚⬚⬚⬚ doing a research study to evaluate the effectiveness of AI assistance for humans to predict adult income. We invite you to take part in this research study because you reside in the US and your native language is English. We expect that you will be in this research study for 2-3 minutes depending on the experiment type. Your participation is voluntary.

**Incentives**: You will be asked 8 main questions. The expected duration is 2-3 minutes. You will get 5 cents for each decision. If you guess correctly, you get 1 additional cents. In the event of an incomplete study, you will NOT receive compensation for answering partial questions.

**PLEASE NOTE: This study contains attention checks to make sure that participants are finishing the tasks honestly and completely.** As long as you read the instructions and complete the tasks with adequate attention, your HIT will be approved. If you fail these checks or finish the HIT hastily, your HIT will be rejected. This study also contains an eligibility check to make sure there are no duplicate participants.

**Risks and Benefits**: Your participation in this study does not involve any risk to you beyond that of everyday life. This study may benefit society by improving the understanding of how AI assistance can improve humans' ability to predict adult income.

**Confidentiality**: Your Worker ID will be used to distribute payment to you. Please be aware that your Worker ID can potentially be linked to information about you on your Prolific public profile page, depending on the settings you have for your Amazon profile. We will not be accessing any personally identifying information about you that you may have put on your platform public profile page.
● If you decide to withdraw, data collected up until the point of withdrawal may still be included in analysis.
● Identifiable data (your worker id) will never be shared outside the research team.
● De-identified information from this study may be used for future research studies or shared with other researchers for future research without your additional informed consent.

**Contacts & Questions**: If you have questions or concerns about the study, you may email questions to ⬚⬚⬚⬚⬚⬚⬚⬚ ⬚⬚⬚⬚⬚⬚⬚⬚⬚⬚⬚⬚

If you have any questions about your rights as a participant in this research, feel you have been harmed, or wish to discuss other study-related concerns with someone who is not part of the research team, you can contact ⬚⬚⬚⬚⬚⬚ ⬚⬚⬚⬚⬚⬚⬚⬚⬚⬚⬚⬚⬚⬚⬚⬚⬚⬚⬚⬚ ⬚⬚⬚⬚⬚⬚⬚⬚⬚.

**Consent**: Participation is voluntary. Refusal to participate or withdrawing from the research will involve no penalty or loss of benefits to which you might otherwise be entitled. By clicking "Agree" below, you confirm that you have read the consent form, are at least 18 years old, and agree to participate in the research. Please print or save a copy of this page for your records. You have not waived any legal rights you otherwise would have as a participant in a research study.

Agree       Do not agree

Figure 11: Consent form.

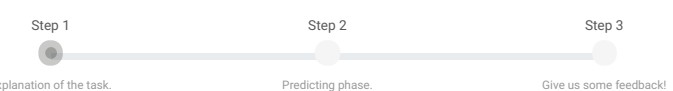

Explanation of the task.  Predicting phase.  Give us some feedback!

In this task, you will go through the **profiles of 8 people**. For each profile, you will **predict whether the person's annual income is above or below $50K.**

Each profile contains two facts about a person:
- **Education** (from middle school to masters). Roughly half of the profiles are above high school (e.g., masters) and half are equal or below (e.g., middle school).
- **Age** (from 20 to 50 years old). Roughly half of the profiles are above 30 and half are below.

**About half of the profiles have an annual income of above $50k.**

An Artificial Intelligence (AI) model will assist you on this task. The AI model uses education, age, and many other factors. For each profile, you will see the AI's prediction and an explanation which points out the key information used to make the prediction. **The AI is not perfect, and hopefully the explanation can help you identify its mistakes.**

It is important that you **complete the exit survey to receive a unique code that is required to be compensated. Please submit the unique code after completing the HIT**.

> Please answer the questions below carefully. The questions test your understanding of this experiment. You will be **disqualified** from the study if any question is answered incorrectly.

**\*1. What is your task in this study?**

- ○ To find the level of education of a person.
- ○ To predict whether a person's income is above 50K or not.
- ○ To find the age of the highest income person.
- ○ To evaluate an AI's performance on income prediction.

**\*2. I have to answer an exit survey after the study to get a unique code, which should be submitted in order to get the compensation.**

- ○ Yes. I agree.
- ○ No. I don't need to submit the code to get compensated.

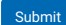 Submit

Figure 12: Task description and attention check for participants in the regular group. The user is required to select the correct answers before they are allowed to proceed to the training phase. The correct answers can be inferred from the given text on this page.

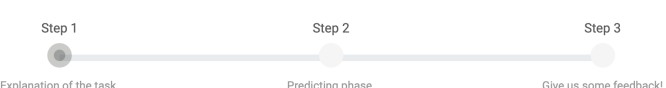

In this task, you will go through the **profiles of 8 people**. For each profile, you will **predict whether the person's annual income is above or below $50K.**

**About half of the profiles have an annual income of above $50k.**

An Artificial Intelligence (AI) model will assist you on this task. The AI model considers education, age, and many other factors. The AI uses a neural network to combine these inputs into **feature A** and **feature B**, which the AI then uses to make the final prediction. You will see the two features, the AI's prediction, and an explanation which points out the feature used to make the prediction. **The AI is not perfect, and hopefully the explanation can help you identify its mistakes.**

It is important that you **complete the exit survey to receive a unique code that is required to be compensated**. **Please submit the unique code after completing the HIT**.

> Please answer the questions below carefully. The questions test your understanding of this experiment. You will be **disqualified** from the study if any question is answered incorrectly.

**\*1. What is your task in this study?**

- ○ To explain the feature A of the AI model.
- ○ To predict whether a person's income is above 50K or not.
- ○ To predict the explanation of a complex neural networks.
- ○ To evaluate an AI's performance on income prediction.

**\*2. I have to answer an exit survey after the study to get a unique code, which should be submitted in order to get the compensation.**

- ○ Yes. I agree.
- ○ No. I don't need to submit the code to get compensated.

**Submit**

Figure 13: Task description and attention check for participants in the anonymized group. The user is required to select the correct answers before they are allowed to proceed to the training phase.

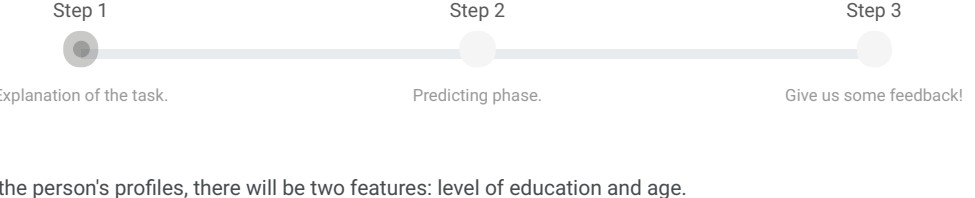

In each of the person's profiles, there will be two features: level of education and age.

**\*Q1. In your opinion, what's the relationship between education and income?**

○  Positively correlated. In other words, people with higher education usually have higher incomes.
○  Negatively correlated. In other words, people with higher education usually have lower incomes.
○  Not correlated.

**\*Q2. In your opinion, what's the relationship between age and income?**

○  Positively correlated. In other words, older people usually have higher incomes.
○  Negatively correlated. In other words, older people usually have lower incomes.
○  Not correlated.

**\*Q3. Which feature do you think is the more indicative of income within the following two?**

○  Education
○  Age

**\* Please rate your agreement with the following statement: I am confident about my answer to the Q3 with respect to which feature is more indicative of income.**

○  Strongly agree
○  Agree
○  Neutral
○  Disagree
○  Strongly disagree

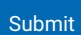

Figure 14: Preliminary questions to calibrate human intuitions. If a participant agrees that both education and age are positively correlated with income and he/she thinks education is more important, the participant holds intuitions consistent with both R and M.

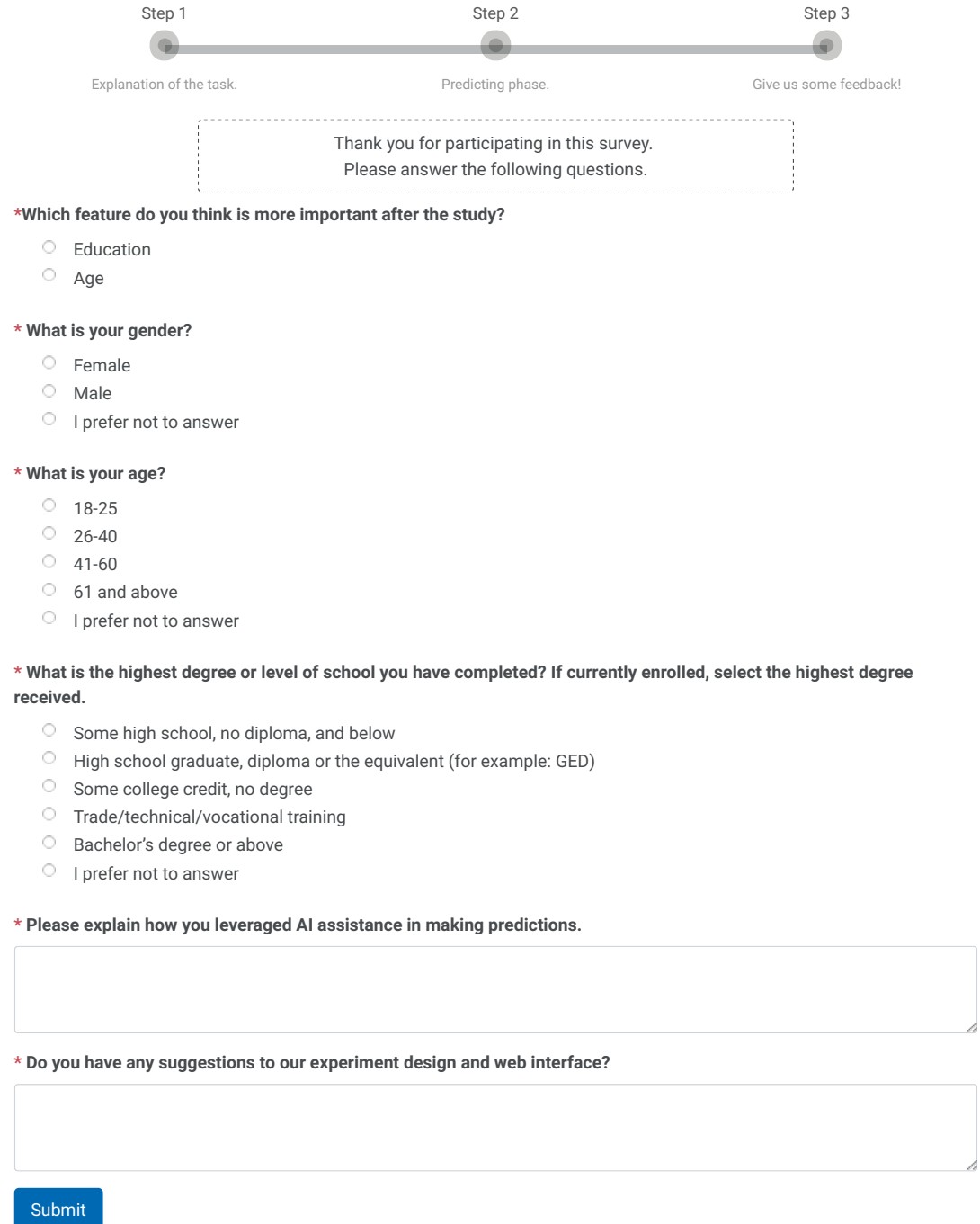

Figure 15: Exit survey.

