# OpenReview forum: "Machine Explanations and Human Understanding"
_TMLR — Accepted by TMLR_

### Review · Reviewer_hQ3d · 2023-01-16

**Summary Of Contributions:**

This submission works towards a theoretical framework for how a person understands a learned model.  To do this, it starts by identifying three core functions that a person may want to understand:  the model boundary, the task boundary, and the model error.   Then, it models the effects of an input, an explanation, and human intuitions on a person's understanding of these core functions using causal graphs and a new operator.  This framework suggests that, without assumptions about these human intuitions, explanations cannot help a person understand the task boundary or the model error.  The submission supports this claim using a simple user study.

**Audience:**

No

**Claims And Evidence:**

No

**Requested Changes:**

The reviewer thinks that the submission needs a substantial re-write to address the above weaknesses.


**Strengths And Weaknesses:**

Strengths:
- Formalizing the three core functions and their interactions seems helpful.
- Incorporating Z into E is an interesting idea that is largely unexplored in work on explainability.

I found many weaknesses with the submission (listed bellow), but would like to make two higher level comments first:
- These weaknesses can be summarized as follows:  this framework treats both "explanations" and "human intuitions" as if they are binary features that are either present or absent and, consequently, it is not clear what the framework tells us about the typical setting where users have some degree of human intuition and we are interested in the effects of different types of explanations.
- While the I am very familiar with explainability, I am not particularly familiar with causality.  So I found the submission fairly difficult to read and may have missed something important.

Weaknesses:
- In the third to last paragraph of Section 1, the authors claim that "Our causal diagrams reveal.... , and shed light on the mixed findings of empirical research on explanations."  This claim is not supported because there is no analysis that looks at whether or not these past empirical findings considered human intuitions.  The example of "deceptive review detection" is interesting, but there should be a more systematic study.
- The framework is based on a novel operator (the show operator) whose affects on the causal graph are defined, but not justified.
    - For example, showing the user $\hat{Y}$ does not seem to be equivalent to showing them $E(X, g)$ because $E$ may contain information about $g$ for points $X'$ that are similar to $X$ when $E$ is a local approximation-based explanation.  This is not the same as considering some notion of "global understanding".
- One of the primary implications of the framework, that "complementary performance is impossible without assumptions about task-specific human intuitions," is not particularly interesting and commonly considered in work on explainability:  if the user has no "task-specific human intuitions" (ie, domain knowledge) it is clear that they cannot help make better predictions.  For example, no one expects a non-expert to be able to make medical decisions because they were shown a model's explanation.
- It is unclear how to extend the framework to model the effects of a specific human intuition rather than simply showing that, without any human intuition, it is impossible to gain certain types of understanding.
- The results of the user study do not support the claims of either Theorem 1 or 2:
    - The participants were asked to make a prediction, so the experiment does not test Theorem 1.
    - There was no measurement presented showing that users made better predictions in the "regular" setting than in the "anonymized" setting.  This result would have supported Theorem 2.
- The results of the user study are not of general interest to the community:
    - The results pertaining to H1 are expected.  The authors note this themselves when they say "For a person who has no intuitions about a task at all, the best course of action is to follow machine predictions."
    - The results pertaining to H2 have been identified by past works.
- The setup considered by the user study is very simplistic.
- The claim that, for discovery tasks, "if we do not make assumptions about human intuitions, E can cannot bring any additional utility over showing $\hat{Y}$" seemingly contradicts the example giving in the "Expanding human intuitions" paragraph.

---

### Review · Reviewer_JmbV · 2023-02-06

**Summary Of Contributions:**

This paper proposes a causal framework for modeling the relationship between ML explanations, based on the idea that human intuitions have a key role in the relationship between humans and understanding explanations. They find that without intuitions surrounding explanations, humans can't understand certain aspects of ML prediction problems, like the task decision boundary or the error of the model. They validate their framework with experimental results using humans.

**Audience:**

Yes

**Claims And Evidence:**

Yes

**Requested Changes:**

Currently, I believe the statement of theorem 2 should be addended to specify that it is applicable for explanations which do not require the use of the predicted label. However, if I'm mistaken, I would greatly appreciate some clarification here.

**Strengths And Weaknesses:**

Overall, I think this is a quite well-written paper. The framework is presented and contextualized well, and it is easy to follow.

After reading the paper, I'm still a bit confused however about the claim in the theorem "Explanations can improve human understanding of model decision boundary, but cannot improve human understanding of task decision boundary or model error." In particular, I'm not entirely sure how explanations couldn't help improve human understanding of the task decision boundary. If I have something like a counterfactual explanation and model fits the task quite well, cfe's should at least provide a noisy approximation of the task decision boundary, even if the features are totally anonymized. I suppose that the claim in the theorem assumes the predicted label is not shown, but it is very hard to separate predicted labels for certain types of explanations, like counterfactual explanations. Perhaps my concern rests with the assumption of the causal model, or this theorem is not applicable to certain types of explanations that are inseparable with the predicted label, like cfe's. Overall, I'm having trouble fully grasping how understanding the task decision boundary from explanations categorically is not true, as it is currently presented in the theorem, and I would love some clarification here.

Minor:
The line: "To the best of our knowledge, we are not aware of any existing quantitative behavioral measure of human understanding that does not belong to one of these three concepts of interest." is a bit confusing. Perhaps this is true  for understanding model/task predictions, but there are other aspects of model understanding (e.g., just understanding the mechanism by which a model makes a prediction) that are not included in these definitions. At least, I think this should be slightly qualified to better suite to problem at hand, and it is a bit confusing to read currently.

---

### Review · Reviewer_SMPN · 2023-03-13

**Summary Of Contributions:**

This paper studies the interplay between human understanding and model explanations, which had not been given a thorough treatment to date. Grounded in an exhaustive search over existing literature, the authors note three factors that govern this relationship: task decision boundary, model decision boundary, and model error. The proposed notion that human intuition is important to explanations is another contribution, which was mere folklore until now. The authors conclude with a human subject experiment of how human intuition impacts a participants understanding of a model explanation.

This is a fine piece of work that I recommend accepting as is.

**Audience:**

Yes

**Broader Impact Concerns:**

No concerns.

**Claims And Evidence:**

Yes

**Requested Changes:**

- Minor: when listing multiple citations for a given fact/statement, please try to ensure the papers are listed in chronological order; this is a stylistic thing but surely improves readability
- If possible, could you scale the human subject experiments to at least 30 instances? This can hopefully permit stronger statements about how participants perceive explanations and how it affects their understanding, as it were.

**Strengths And Weaknesses:**

Strengths
- The paper poses an interesting framing of how model explanations help user understanding of ML models.
- Section 2 is thorough and well-written.
- Table 2 (and much of the Appendix) is a very helpful resource for the explanation community.

Weaknesses
- None come to mind.
- I would have preferred a more robust human subject experiment. Since only 16 instances were used, I'm not sure we can safely conclude much (even though the number of participants was more than sufficient).

---

### Decision · Action_Editors · 2023-04-04

**Recommendation:** Accept with minor revision

**Comment:**

Overall, the proposed framework and a focus on the role of human intuition in the effectiveness of explanation are useful notions and the manuscript is well constructed. As said above, I believe there is an audience for this work and that the claims could be adjusted to better align with the presented results. I am recommending acceptance with a minor revision. In the revision, I would like to see author adjust claims on how their work can be used to shed light on prior studies and how the human studies validate elements of their proposed framework. I would also like the bolded claim at the top of page 8 to more clearly include the assumed lack of human intuition.

**Audience:**

All reviewers agree there is an audience for this work in the TMLR community. Reviewer hQ3d expressed some concerns that the results may be unsurprising or that the framework may need extension to be able to handle more realistic notions of human intuition. The AE believes there exists an audience that will appreciate the initial framework proposed here and the accompanying human studies and supplemental material.

**Claims And Evidence:**

Reviewer hQ3d questions whether this work sufficiently supports the claim that the proposed framework "shed[s] light on the mixed findings of empirical research on explanations". While the proposed theory would suggest prior studies' results are influenced by the presence or absence of task-relevant human intuitions, the AE agrees with hQ3d that this has not been evaluated with a thorough accounting of the problem settings in prior work. Authors extend their argument in the response, describing some prior work and how the proposed theoretical framework might explain the reported results.


Reviewer hQ3d also questioned whether the user study sufficiently supports the proposed framework. The two theorems presented are meant to describe how explanations require human intuition in order to improve human understanding of task decision boundary or model error. I believe Theorem 2 could have been validated with the proposed experiments if the anonymized setting included conditions with and without explanation, then measured differences in user-model agreement. But this was not done. The remaining results focus on the prediction-given setting with intuition, which does not cover the presented theorems. However, the framework is more than the theorems and the human study does examine the relationships between human intuition and explanation described by the framework.

The AE believe these issues could be resolved by greater specificity about the claims. For instance, explaining how and which parts of the framework the human studies validate when making related claims. Likewise, being specific about how the proposed framework might be used to explain prior work rather than saying it directly sheds light on prior results.

Neither other reviewer raises concerns about the support of main claims of this work.